Comparison of soil quality assessment methods for different vegetation eco-restoration techniques at engineering disturbed areas

Zhao Bingqin 1 2
Gao Ruzhang 1 2
Zhang Xingfeng 1 2
Xia Lu 1 2
Zhang Lun 1 2
Xia Dong 1 2 3
Liu Daxiang 1 2
Xia Zhenyao 1 2 xzy_yc@126.com
Xu Wennian 1 2 3
1 Key Laboratory of Geological Hazards on Three Gorges Reservoir Area, Ministry of Education, China Three Gorges University , Yichang , China
2 Hubei Key Laboratory of Disaster Prevention and Mitigation, China Three Gorges University , Yichang , China
3 Hubei Provincial Engineering Research Center of Slope Habitat Construction Technique Using Cement-based Materials, China Three Gorges University , Yichang , China
Mahmood Haider
Electronic publication date: 2024 Sep 5
Publication date: 2024
Volume: 12
Electronic Location ID: e18033
Received 2024 Mar 1; Accepted 2024 Aug 12
Copyright: © 2024 Zhao et al.
Copyright year: 2024
Copyright holder: Zhao et al.
License: This is an open access article distributed under the terms of the Creative Commons Attribution License, which permits unrestricted use, distribution, reproduction and adaptation in any medium and for any purpose provided that it is properly attributed. For attribution, the original author(s), title, publication source (PeerJ) and either DOI or URL of the article must be cited.
License URL: https://creativecommons.org/licenses/by/4.0/

Keywords: Engineering disturbed areas, Vegetation eco-restoration, Soil quality indicators, Methods comparison

Funding: Open Fund of Key Laboratory of Geological Hazards on Three Gorges Reservoir Area (China Three Gorges University) Ministry of Education 2023KDZ18 National Natural Science Foundation of China 52200230 This work was supported by the Open Fund of Key Laboratory of Geological Hazards on Three Gorges Reservoir Area (China Three Gorges University), the Ministry of Education (2023KDZ18), the National Natural Science Foundation of China (52200230). The funders had no role in study design, data collection and analysis, decision to publish, or preparation of the manuscript.

==============================
Scientific assessment of soil quality is the foundation of sustainable vegetation eco-restoration in engineering disturbed areas. This study aimed to find a qualitative and comprehensive method for assessing soil quality after vegetation eco-restoration in engineering disturbed areas. Sixteen soil indicators were used at six vegetation eco-restoration sites as the potential soil indicators. A minimum data set (MDS) and revised minimum data set (RMDS) were determined by principal component analysis. Six soil quality indices (SQIs) of varying scoring functions based on different data sets were employed in this study. Significant positive correlations were observed among all six SQIs, indicating that the effects of different vegetation eco-restoration measures on soil quality could be quantified by all six SQIs. The SQI values of the vegetation concrete eco-restoration slope (VC), frame beam filling soil slope (FB), thick layer base material spraying slope (TB), and external-soil spray seeding slope (SS) were all significantly higher than the SQI value of the abandoned slag slope (AS). It is noteworthy that the SQIs of the VC and TB sites were also significantly higher than the SQI of the natural forest (NF) site. These results indicate that the application of artificial remediation measures can significantly improve the soil quality of the disturbed area at the Xiangjiaba hydropower station. The results of this study also indicate that the SQI-NLRM method is a practical and accurate quantitative tool for soil quality assessment and is recommended for evaluating soil quality under various vegetation eco-restoration techniques in disturbance areas at the Xiangjiaba hydropower station and in other areas with similar habitat characteristics.

Introduction

Soil quality is defined as “the capacity of soil to function within ecosystem and land use boundaries, to sustain productivity, maintain environmental quality, and to promote plant growth as well as animal health” (Doran & Parkin, 1994; Neri et al., 2023). Indicators of soil quality generally include physical, chemical, and biological properties that may collectively or individual contribute to the accretion, maintenance, or degradation of soil functions under various land uses and management practices (Persico et al., 2023). Many studies have been published by previous researchers on methods of assessing soil quality (Table 1). Celik et al. (2021) found that available phosphorus and potassium concentrations, plant available water capacity, penetration resistance, potentially mineralizable nitrogen, microbial biomass carbon, and soil pH were all important indicators for evaluating and quantifying the impacts of long-term conservation and conventional tillage practices used in wheat-corn-soybean rotation. Yang et al. (2021) used electrical conductivity, pH value, main cation ions content, and available nutrients to investigate the influence of various Tamarix chinensis-grass patterns on coastal soil quality. Yang et al. (2023) proposed that the carbon source provided by soil played a vital role in the formation of aggregates in cultivated land with different fertilization treatments, and could be used to characterize the dominant factors of soil structure formation. Yu et al. (2017) found that soluble organic carbon, active organic carbon, invertase, and the N/P ratio were able to characterize changes in soil quality under different land use patterns in northern China. Raiesi (2017) found that soil organic carbon, electrical conductivity, and arylsulfatase were important indicators for evaluating soil quality degradation after cultivation of the land in arid and semi-arid upland surroundings. Wang et al. (2023b) used nine soil parameters for the soil quality evaluation of grassland in the Qinghai–Tibet Plateau: soil water content, soil bulk density, capillary porosity, non–capillary porosity, soil organic matter, total nitrogen content, total phosphorus content, total potassium content, and alkali–hydrolyzable nitrogen. The evaluation of soil quality is limited by multiple factors including changes in specific soil types and regional environmental factors (such as temperature, rainfall, etc.). Different forms of land use also lead to changes in soil indicators, thus affecting the results of soil quality evaluations. Combining various soil properties into an integrated index may create a better measure of soil quality than individual indicators (Li et al., 2023).

Table 1 Summary of soil quality assessment methods.

Soil assessment methods	Research areas	Indicator	Summarization	
Celik et al. (2021)	Soil quality of long-term conservation and conventional tillage practices used in wheat-corn-soybean rotation	Available phosphorus and potassium concentrations, plant available water capacity, penetration resistance, potentially mineralizable nitrogen, microbial biomass carbon, and soil pH	Combining various soil properties into an integrated index can assess soil quality better than individual indicators.

No single method can be applied to all soil quality assessments due to the inherent differences between soils. The comprehensive evaluation of soil quality in a specific region must be carried out on the premise of fully understanding the regional soil characteristics

	
Yang et al. (2021)	Soil quality of various Tamarix chinensis-grass patterns on coastal soil quality	Electrical conductivity, pH value, main cation ions contents, and available nutrients	
Yang et al. (2023)	Soil structure of cultivated land	Carbon source indicators	
Yu et al. (2017)	Soil quality under different land use patterns in northern China	Soluble organic carbon, active organic carbon, invertase, and the N/P ratio	
Raiesi (2017)	Soil quality degradation after converting to cultivated land in arid and semi-arid upland surroundings	Soil organic carbon, electrical conductivity, and arylsulfatase	
Wang et al. (2023b)	Soil quality evaluation of grassland in the Qinghai–Tibet Plateau	Soil water content, soil bulk density, capillary porosity, non–capillary porosity, soil organic matter, total nitrogen content, total phosphorus content, total potassium content, alkali–hydrolyzable nitrogen	
Zhang et al. (2022)	Soil quality under various composting treatments	The ‘S’ type membership function	

Different soil indicators often depend on each other, scholars began to pay attention to how to conduct an efficient and accurate evaluation of soil quality, and tried to find a systematic method to measure and explain soil characteristics (Qian et al., 2023). The introduction of the principle of fuzzy mathematics has led to the proposal of several methods that improve upon the scoring standard and weight determination methods, including: the comprehensive index method, the relative soil quality index method, the fuzzy clustering evaluation method, the analytic hierarchy process, the distance between the superior and inferior solutions, the grey clustering method, the neural network method, the linear regression method, the grey correlation degree method, and the minimum data set method (Raiesi, 2017; Yang et al., 2021; Qian et al., 2023; Zhang et al., 2023c, 2023b). The keys to creating an index method capable of comprehensively evaluating soil quality are the selection of soil indicators, the index scoring, and weight determination. It has been suggested that both linear and nonlinear membership functions can reflect the change impacts of land utilization type on soil quality and deterioration in the natural pasture of highland arid and semi-arid areas (Raiesi, 2017). Zhang et al. (2022) proposed that the ‘S’ type membership function can be applied to the comprehensive appraisal of farmland soil quality under various composting treatments. It is worth noting that no single method can be applied to all soil quality assessments due to the inherent differences between soils (Yang, Qiao & Li, 2023). A complete understanding of specific regional soil characteristics is required to establish a soil quality evaluation index system suitable for each specific region and the specific purpose of each soil evaluation (Raiesi & Pejman, 2021; Tang et al., 2023).

Construction, mining, and hydraulic projects cause strong disturbances to the ecosystem by changing surface structure and causing large-scale vegetation destruction (Shen et al., 2023). For example, at the Xiangjiaba hydropower station in the upper reaches of the Yangtze River, the disturbed area accounts for more than 50% of the total construction area (Zhao et al., 2021). The backfill from engineering excavation and overburden stripping forms loose bare slopes, which are the main factors in the degradation of slope ecosystems, affecting the ecological environment and landscape of the project area (Wang et al., 2023a). Vegetation restoration is an effective ecological compensation measure that balances the health of the ecological environment with social and economic development (Shen et al., 2023). Vegetation restoration can both improve the soil quality of engineering disturbed areas and restore the soil ecosystem (Wan et al., 2023). Scholars have conducted a wide range of fundamental studies on the vegetation restoration of artificial disturbed areas and a series of advancements have been made (Shen et al., 2023; Xu et al., 2023). Most studies on the topic focus on vegetation eco-restoration technology selection, the law of vegetation succession, or variation of soil characteristics after vegetation restoration of engineering disturbed areas, but more studies are needed on soil quality evaluation under different vegetation eco-restoration techniques (Hu et al., 2021; Thakur et al., 2022). Scientific evaluation of soil quality is the basis for understanding ecological restoration. The ecological restoration outcomes of different restoration techniques in disturbed areas are assessed by comparing soil quality. The establishment of an evaluation index system for quantitative soil quality assessment would be especially useful for assessing the effects of different vegetation eco-restoration techniques at engineering disturbed areas.

This study was conducted with the following objectives: (1) determine a data set for soil quality in different vegetation eco-restoration techniques at the Xiangjiaba hydropower station; (2) develop a soil quality index (SQI) using linear and nonlinear scoring functions for various vegetation eco-restoration techniques at the Xiangjiaba hydropower station; and (3) determine the effects of various vegetation eco-restoration techniques on the soil quality of disturbed areas at the Xiangjiaba hydropower station.

Materials and Methods

Study area and experimental design

This study was conducted at the Xiangjiaba hydropower station, which is located at the convergence of the Sichuan and Yunnan provinces, in southeast China (Fig. 1). The experimental sites are positioned within the major prevention area of soil and water conservation zone along the upper reaches of the Yangtze River. The area is characterized by a subtropical monsoon climate with an annual average precipitation of 1,078 mm, with approximately 90% of the total precipitation occurring between May and October. The annual average temperature ranges from 12.0 °C to 18.1 °C. Sample sites were chosen with similar evolutionary times, but with varied common eco-restoration techniques. Four vegetative ecological restoration techniques were selected to determine which soil quality assessment method was more applicable to the disturbed slopes in the area: vegetation concrete eco-restoration slope (VC), frame beam filling soil slope (FB), thick layer base material spraying slope (TB), and external-soil spray seeding slope (SS). For comparison, the soil samples from the abandoned slag slope (AS) and the undisturbed, natural forest site (NF) were used to assess whether vegetation eco-restoration techniques had an effect on soil quality. Every sample location had planting soils from the Xiangjiaba Disturbed Area, and all of the sample sites had nearly identical climates. A brief description of the study sites is presented in Table 2.

Figure 1 The location of the study area.

Vegetation concrete eco-restoration slope (VC), frame beam filling soil slope (FB), thick layer base material spraying slope (TB), external-soil spray seeding slope (SS), abandoned slag slope (AS), and natural forest (NF). Satellite image: Google Earth Image ©2024 CNES/Airbus. Photo credit: Xia Dong.

Table 2 Brief descriptions of the study sites.

Site	Technique	Latitude	Longitude	Altitude
(m)	Slope gradient (°)	Eco-restoration time	
VC	Vegetation concrete eco-restoration technique	28°38′N	104°24′E	328.50	63	2004.12	
FB	Frame beam filling soil technique	28°38′N	104°24′E	288.9	40	2004.11	
TB	Thick layer base material spraying technique	28°38′N	104°26′E	388.9	51	2004.12	
SS	External-soil spray seeding technique	28°39′N	104°23′E	473.9	30	2005.06	
AS	Abandoned slag slope	28°38′N	104°24′E	520.5	42		
NF	Natural forest	28°39′N	104°23′E	502.4	45		

Soil sampling and laboratory analyses

Soil samples were collected at a 0–20 cm depth with a 5 cm diameter soil sampler. Five sample plots (1 m × 1 m) were chosen from randomly selected locations to represent each vegetation eco-restoration technique, with each sample plot spaced at least 6 m apart and 1 m from the site edge (Ma et al., 2022). Visible debris, such as root materials and organic residues, was removed from each soil sample before dividing the soil sample into two sub-samples: one subsample for the microbial properties analysis that was stored at 4 °C, and one for the physical and chemical properties analysis that was air-dried and then passed through a 2 mm sieve. The following indicators were measured based on existing soil characteristic analyses of the study area (Xu et al., 2017): soil bulk density (BD), soil water content (WAT), soil porosity (POR), soil organic matter (SOM), total nitrogen (TN), available nitrogen (AN), total potassium (TK), available potassium (AK), pH, urease (URE), catalase (CAT), invertase (INV), polyphenol oxidase (PPO), microbial biomass carbon (MBC), microbial biomass nitrogen (MBN), and microbial biomass phosphorus (MBP). The total of sixteen soil indicators were measured using corresponding standard analytical methods (refer to Table 3 for details).

Table 3 Analytical methods for measuring soil indicators.

Soil process	Soil indicator	Abbreviation	Methods	Reference	
Physical	Soil bulk density	BD	Core method	Clancy et al. (2023)	
Soil water content	WAT	Gravimetric with oven drying method	Almutawa & Eid (2023)	
Soil porosity	POR	Core method	Manoj, Thirumurugan & Elango (2020)	
Chemical	Soil organic matter	SOM	Potassium dichromate titrimetric method	Li et al. (2021)	
Total nitrogen	TN	Spectrophotometer method based on the modified Berthelot reaction with an Skalar San++ continuously flowing autoanalyzer	Page, Miller & Kenney (1982), Zhang et al. (2006)	
Available nitrogen	AN	
Total potassium	TK	
Available potassium	AK	
pH	pH	Electrometric method	Liu et al. (2018)	
Biological	Urease	URE	10% urea, pH = 6.7, 37 °C, 24 h	Yu et al. (2017)	
Catalase	CAT	Permanganate titration	Johnson & Temple (1964)	
Invertase	INV	3,5-dinitro salicylic acid colorimetry, pH = 5.5, 37 °C, 24 h	Yu et al. (2017)	
Polyphenol oxidase	PPO	Pyrogallol colorimetry, pH = 4.5, 30 °C 2 h	Dogan & Dogan (2004)	
Microbial biomass carbon	MBC	Chloroform fumigation technique	Vance, Brookes & Jenkinson (1987), Brookes et al. (1985), Brookes, Powlson & Jenkinson (1982)	
Microbial biomass nitrogen	MBN	
Microbial biomass phosphorus	MBP	

Soil quality assessment method

A principal component analysis (PCA) was performed to select the potential soil indicators that best represented soil functions and processes from the total data set (TDS). The minimum data set (MDS) and the revised minimum data set (RMDS) were identified by selecting factors that had an eigenvalue greater than 1 and accounted for 5% or more of the variance in the database. For each principal component (PC), indicators received a weight representing their contributions to the PC. The selected indicators included in MDS and RMDS were those with a loading value within 10% range of the maximum weighted loading of each PC. A multivariate correlation analysis was conducted when there was more than one soil indicator contained in each PC (Raiesi & Pejman, 2021). All soil indicators with no correlation found between them were retained in the MDS or RMDS. Well-correlated soil indicators were considered surplus and only the highest weighted indicator was chosen for the MDS or RMDS.

Two soil scoring functions, including linear and nonlinear membership functions, were selected to transform soil indicators in MDS into unitless scores ranging from 0 to 1. Each soil indicator’s effect on soil quality was scored in an increasing order by a “more is better” scoring curve and in decreasing order by a “less is better” scoring curve. The following equations were used to compute linear scoring curves:

(1) SL=X/Xmax

(2) SL=X/Xmin

where SL is the linear score ranging from 0 to 1; X is the value of the soil indicator; and xmax and xmin are the maximum and minimum of each soil indicator value, respectively, corresponding to the soils of the six sampling sites (Raiesi, 2017). The “more is better” scoring was computed by Eq. (1) and the “less is better” scoring was computed by Eq. (2). The following equation was used for nonlinear scoring:

(3) SNL=A/(1+(x/x¯)B)

where SNL is the nonlinear score of the soil indicator ranging from 0 to 1; x is the soil indicator value; x¯ is the mean value of each soil indicator observed among the six sampling sites; A is defined as the maximum score of SL, which is 1 in this study; and B is defined as the slope of the equation, which is set as +2.5 for the “less is better” curves and −2.5 for the “more is better” curves (Raiesi, 2017; Yu et al., 2018; Wang et al., 2023b).

The values of the selected indicators were transformed into a comparative SQI through the following weighted additive (Eq. (4)) equation:

(4) SQI=∑i=1nwi×Si

where SQI is the weighted additive of the soil quality indexes, ranging from 0 to 1; n is the soil indicator quantity included in the data set; wi is the weight of the soil indicator; and Si is the score value (Qian et al., 2023).

Six evaluation methods were determined by various scoring functions: linear scoring based on the total data set (SQI-LT), nonlinear scoring based on the total data set (SQI-NLT), linear scoring based on the minimum data set (SQI-LM), nonlinear scoring based on the minimum data set (SQI-NLM), linear scoring based on the revised minimum data set (SQI-LRM), and nonlinear scoring based on the revised minimum data set (SQI-NLRM). The SQI reflects the effects of vegetation eco-restoration techniques on soil function and soil process (Raiesi, 2017; Yu et al., 2018), with a higher SQI value indicating better soil function. Soil quality grades were compared and a correlation analysis was conducted to estimate the effect of different ecological restoration techniques.

Statistical analysis

All statistical analyses were conducted with Statistical Product and Service Solutions (SPSS 20.0) software. The effects of soil indictors and SQIs were analyzed using one-way analysis of variance (ANOVA) and means were compared with Fisher’s least significant difference test (LSD). The relativity of soil indicators and SQIs was analyzed using Pearson correlation coefficients. Principal component analysis (PCA) was used to determine indicators included in MDS or RMDS and calculate the weights of soil indicators based on norm values.

Results

Changes in soil quality indicators

A total of sixteen soil indicators were measured at the Xiangjiaba hydropower station as potential indicators of soil quality under different vegetation eco-restoration techniques (Table 4). Different vegetation eco-restoration techniques had different effects on soil indicators. Soil originating from the VC site had significantly higher levels of AK, URE, and MBP, and the lowest level of MBN (P < 0.05) compared to the soil from other sites. The levels of soil indicators, including TN, MBC, and MBP, at the FB site were significantly lower than those at other sites (P < 0.05). Values of SOM, TN, CAT, and INV were highest at the TB site. Soil originating from the AS site had the lowest values of WAT, POR, SOM, AN, URE, INV, and PPO, but significantly higher levels of BD, TK, pH, and MBN than the soil from other sites (P < 0.05). Soil from the NF site had significantly higher levels of WAT, POR, AN, and MBC, and significantly lower levels of BD, TK, pH, and CAT (P < 0.05) than the soil from other sites. The coefficient of variation (CV) had a minimum value of 12.21% (pH) and a maximum value of 84.35% (URE).

Table 4 Measurements of soil indicators at each study site.

Indicators	VC	FB	TB	SS	AS	NF	CV (%)	
BD (g cm−3)	1.09 ± 0.14bc	1.35 ± 0.02a	1.07 ± 0.05bc	1.12 ± 0.09b	1.54 ± 0.11a	0.87 ± 0.19c	20.61	
WAT (%)	37.24 ± 6.07ab	19.80 ± 1.12cd	35.27 ± 6.71ab	29.82 ± 0.79bc	8.60 ± 1.24d	46.66 ± 15.27a	47.78	
POR (%)	58.94 ± 5.26ab	48.99 ± 0.82c	59.73 ± 1.69ab	57.85 ± 3.18b	41.81 ± 4.25c	67.11 ± 7.55a	16.46	
SOM (g kg−1)	23.73 ± 0.62b	12.13 ± 0.25c	27.91 ± 0.71a	10.45 ± 0.24d	6.36 ± 0.46e	27.09 ± 0.35a	49.19	
TN (g kg−1)	1.72 ± 0.03c	1.08 ± 0.05d	2.70 ± 0.21a	1.08 ± 0.10d	1.11 ± 0.09d	2.18 ± 0.28b	39.83	
AN (mg kg−1)	94.30 ± 8.08b	20.09 ± 0.48d	111.28 ± 0.39a	30.90 ± 2.37c	10.66 ± 0.57e	115.72 ± 0.47a	71.33	
TK (g kg−1)	18.76 ± 0.34c	19.54 ± 0.31b	19.80 ± 0.25b	17.01 ± 0.57d	28.11 ± 0.37a	15.20 ± 0.22e	21.24	
AK (mg kg−1)	405.76 ± 18.14a	176.99 ± 5.35e	266.22 ± 8.86c	83.22 ± 1.73f	219.74 ± 18.93d	290.47 ± 12.59b	42.91	
pH	6.34 ± 0.49b	6.81 ± 0.09a	6.03 ± 0.20b	6.12 ± 0.17b	7.06 ± 0.09a	4.84 ± 0.06c	12.21	
URE (mg g−1 d−1)	1.42 ± 0.10a	0.26 ± 0.02d	1.19 ± 0.01b	0.27 ± 0.003d	0.12 ± 0.02e	0.41 ± 0.02c	84.35	
CAT (mL g−1)	6.65 ± 0.06b	6.40 ± 0.04c	6.98 ± 0.07a	6.78 ± 0.12b	4.21 ± 0.01d	3.40 ± 0.09e	25.05	
INV (mg g−1 d−1)	22.30 ± 0.25c	47.29 ± 2.45b	59.54 ± 0.25a	49.37 ± 1.59b	9.58 ± 0.02d	10.29 ± 1.12d	61.75	
PPO (mg g−1 d−1)	54.16 ± 0.21b	44.61 ± 3.25c	33.41 ± 0.73d	61.18 ± 5.14a	17.06 ± 0.05e	21.39 ± 0.50e	43.50	
MBC (mg kg−1)	232.65 ± 10.64c	184.89 ± 12.01d	408.55 ± 8.55b	195.21 ± 4.93cd	377.69 ± 38.39b	455.69 ± 29.69a	36.50	
MBN (mg kg−1)	5.16 ± 0.09e	6.01 ± 1.72e	18.85 ± 0.05c	21.93 ± 2.26b	44.21 ± 1.10a	9.77 ± 0.79d	78.39	
MBP (mg kg−1)	16.43 ± 0.48a	4.40 ± 0.14d	10.70 ± 0.43c	13.74 ± 0.85b	16.37 ± 1.23a	5.00 ± 0.33d	45.90	
Note:

Abbreviations for soil indicators and ecological restoration techniques are shown in Tables 2 and 3, different letters refer to significant differences at 0.05 level.

Soil quality index

Table 5 shows the principal component analysis results of all indicators, the soil indicators differed significantly by vegetation eco-restoration technique across the study sites. The four principal components with eigenvalues >1 explained more than 91% of the variability in the original data (Table 5). These four indicators also explained more than 90% of the variance in SOM, TN, AN, TK, pH, URE, CAT, INV, PPO, and MBC, 70% of the variance in BD, WAT, POR, AK, and MBN, and 66.5% of the variance in MBP. The variability of most soil characteristics were well described by these four components. AN had the highest loading value (0.951), followed by SOM (0.950), WAT (0.920), BD (−0.918), and POR (0.916). Loading values of AN were within 10% of the maximum weighted loading of PC1, which accounted for 47.485% of the total variability (Fig. 2). Due to the significant correlation between AN, SOM, WAT, BD, and POR (P < 0.01; Fig. 3), AN was the only variable included in the MDS and reserved for the SQI estimation. PC2 explained 22.999% of the total variance and was highly loaded with PPO (0.942) and CAT (0.892; Table 5 and Fig. 2). PPO was significantly correlated with CAT (P < 0.01; Fig. 3), but had the highest loading value of the two indicators, so only PPO was retained as the indictor of PC2. PC3 explained 13.882% of the total variance and was highly loaded with URE (0.753) and MBP (0.684; Table 5). No significant differences were found for URE and MBP, so both were selected as indicators of the minimum data set. PC4 explained 7.432% of the total variance, and INV was the only highly loaded indicator (0.661; Table 5). The final indicators retained in the MDS were AN, PPO, URE, MBP, and INV.

Table 5 An analysis of principal components reflecting soil indicators.

Soil indicators	Total data set	Biological	Chemical	Physical	
PC1	PC2	PC3	PC4	COM	PC1	PC2	PC3	COM	PC1	PC2	Communalities	PC1	C	
BD	−0.918	−0.086	0.201	0.089	0.898								−0.993	0.985	
WAT	0.920	0.066	−0.120	−0.143	0.885								0.992	0.985	
POR	0.916	0.087	−0.204	−0.089	0.896								0.968	0.938	
SOM	0.950	−0.086	0.227	0.124	0.977					0.983	0.108	0.979			
TN	0.802	−0.239	0.290	0.435	0.973					0.881	0.210	0.821			
AN	0.9 51	−0.163	0.227	0.069	0.987					0.991	0.097	0.991			
TK	−0.758	−0.419	0.432	0.126	0.952					0.624	0.672	0.866			
AK	0.534	−0.286	0.631	−0.356	0.892					−0.645	0.671	0.841			
pH	−0.832	0.222	0.404	0.046	0.906					−0.803	0.466	0.861			
URE	0.599	0.257	0.7 53	−0.004	0.993	0.467	0.079	0.8 76	0.992						
CAT	0.005	0.892	0.362	0.254	0.991	0.946	0.171	0.048	0.927						
INV	0.051	0.744	−0.013	0.661	0.994	0.748	−0.133	−0.141	0.597						
PPO	0.039	0.942	0.033	−0.252	0.954	0.914	0.113	−0.164	0.876						
MBC	0.388	−0.845	0.047	0.316	0.966	−0.763	−0.126	0.488	0.837						
MBN	−0.660	−0.496	0.140	0.236	0.757	−0.539	0.702	−0.291	0.868						
MBP	−0.345	0.016	0.684	−0.279	0.665	0.082	0.964	0.196	0.974						
Eigenvalue	7.598	3.680	2.221	1.189		3.388	1.504	1.178		4.175	1.183		2.908		
Variance (%)	47.485	22.999	13.882	7.432		48.397	21.488	16.382		69.583	19.717		96.932		
Cumulative variance (%)	47.485	70.485	84.366	91.799		48.397	69.885	86.717		69.583	89.300		96.932		
Note:

Abbreviations for soil indicators and ecological restoration techniques are shown in Tables 2 and 3; PC, principal component; boldface factor loading values are considered highly weighted. Boldface and underlined loading values correspond to the soil indicators included in the MDS.

Figure 2 Principal component analysis plot of soil quality indicators.

Abbreviations for soil indicators are shown in Table 3.

Figure 3 Correlation matrix for measured soil indicators.

Two asterisks (**) indicate significant difference at P < 0.01, one asterisk (*) indicates significant difference at P < 0.05.

Similar to the TDS, the principal component analysis results of biological properties showed that three PCs were identified with eigenvalues >1 and explained more than 86% of the variability of the soil biological data (Table 5). The two highly loaded indicators under PC1, CAT and PPO, were significantly correlated with each other (Fig. 3), so CAT was selected to represent PC1 because it had the highest loading value of the two (0.946). Only MBP (0.964) and URE (0.876) were highly loaded indicators in PC2 and PC3, respectively. For chemical properties, two PCs were identified with eigenvalues >1 and explained more than 89% of the variability of the soil chemical data (Table 5). The two highly loaded indicators under PC1, AN and SOM, were significantly correlated with each other (Fig. 3), so AN was selected to represent PC1 because it had the highest loading value of the two (0.991). TK (0.672) and AK (0.671) were the two highly loaded indicators under PC2, and were both selected as indicators for the revised minimum data set. The principal component analysis results of soil physical property data showed that only one PC was retained for physical properties. All the measured soil indicators in physical properties were significantly correlated with each other (Fig. 3). Therefore, only BD (−0.993) was included in the revised minimum data set. As a result, CAT, MBP, URE, AN, TK, AK, and BD were chosen to establish the revised minimum data set.

The TDS, MDS, and RMDS soil indicators were transformed using the structurally based, linear and non-linear membership functions (Eqs. (1–3)). The type of scoring curve used was determined by the contribution of the indicator to soil function. The weight of each soil indicator was determined using the ratio of its communality to the sum of the communalities of all soil indicators in the TDS or MDS. To establish the RMDS, the soil physical, chemical, and biological properties were first given an equal weight value (0.333) to emphasize the equal importance of these three types of soil properties to the soil ecosystem (Nakajima, Lal & Jiang, 2015; Yu et al., 2023). Then, a sub-weight value of each soil indicator to soil physical, chemical, and biological properties was determined by the ratio of soil indicator’s communality to the sum of the communalities of that specific type of soil property (Yu et al., 2018). The final soil quality index for TDS, MDS, and RMDS can be described as follows:

SQI-M = 0.215 SAN + 0.208 Sppo + 0.216 SURE + 0.145 SMBP + 0.216 SINV

SQI-RM = 0.107 SCAT + 0.112 SMBP + 0.114 SURE + 0.122 SAN + 0.107 STK + 0.104 SAK + 0.333 SBD

SQI-T = 0.061 SBD + 0.060 SWAT + 0.061 SPOR + 0.067 SSOM + 0.066 STN + 0.067 SAN + 0.065 STK + 0.061 SAK + 0.062 SpH + 0.068 SURE + 0.067 SCAT + 0.068 SINV + 0.065 SPPO + 0.066 SMBC + 0.052 SMBN + 0.045 SMBP

Changes in soil quality index

The SQI values under different vegetation eco-restoration measures varied from 0.37 to 0.77 for SQI-LT and from 0.23 to 0.63 for SQI-NLT in the site order of TB > VC > NF > FB > SS > AS (Fig. 4). The SQI value of the AS site was significantly lower than that of the other sites. The SQI values of the artificial vegetation eco-restoration sites were 48% to 106% higher for SQI-LT and 78% to 174% higher for SQI-NLT than the SQI values of the AS site. These results indicated that the artificial vegetation eco-restoration measures applied in the Xiangjiaba hydropower station have markedly improved soil quality. The range of the SQI values across the sites was 0.16 to 0.76 for SQI-LM, 0.07 to 0.70 for SQI-NLM, 0.38 to 0.81 for SQI-LRM, and 0.25 to 0.64 for SQI-NLRM. The positive relationships (Fig. 5) among the six SQIs indicate that all six SQIs can both sensitively and accurately represent and quantify the effects of different vegetation eco-restoration measures on soil quality. The regression analysis between SQI values based on the data sets of different soil indicators (Fig. 6) showed that both linear and nonlinear scoring functions yielded similar SQI values based on MDS and RMDS compared to SQI values based on TDS, and a high correlation was observed between TDS and RMDS for both the nonlinear and linear scoring functions (Figs. 5, 6).

Figure 4 The soil quality indices of different vegetation eco-restoration techniques.

Bars with different letters within vegetation eco-restoration techniques refer to significant differences at the P < 0.05 level.

Figure 5 Correlation of the six soil quality indices.

Two asterisks (**) indicate significant difference at P < 0.01.

Figure 6 Regression analysis of SQI based on the data set of different soil indicators.

(A) Regression between SQI based on TDS and MDS; (B) regression between SQI based on TDS and RMDS.

Discussion

Effect of vegetation eco-restoration techniques on soil quality indicators

The responses of soil quality indicators to different vegetation eco-restoration techniques were not consistent (Table 4), and contradictory results suggest that different restoration measures have complex influences on soil indicators (Celik et al., 2021). The highest WAT value was found in the soil from the NF site, while the lowest WAT value was found in the soil from the AS site (Table 4). This is probably due to difference in species composition at different sites. The dominant species of the NF and FB sites were trees, whose canopy can effectively shield the soil from solar radiation and reduce water evaporation. The vegetation community species richness and vegetation density at the VC and SS sites were higher than those the AS site, leading to good surface water retention at the VC and SS sites, but low water retention and storage capacity at the AS site. In contrast to the AS site, the artificial vegetation eco-restoration sites had significantly increased SOM values (Table 4). According to the classification criterion of soil nutrients, the SOM levels at the artificial vegetation eco-restoration sites were at medium levels or slightly above, but the AS site was considered SOM-deficient (Xu et al., 2017). Artificial disturbance has a negative effect on SOM, while artificial vegetation eco-restoration techniques have a positive effect on SOM. Targeted artificial control measures and improved soil management levels can effectively promote the process of vegetation restoration in disturbed areas (Xu et al., 2017).

The URE values of the VC and TB sites were significantly higher than those at other sites (Table 4). This is mainly because the dominant species of these two sites belong to Leguminosae, and thus the soil of these two sites has strong nitrogen availability. Many studies show a positive relationship between soil nitrogen supply capacity and urease activity (Jin et al., 2009; Jahangir et al., 2021; Zeng, Perry & Toyota, 2023). The lower AN value at the AS site was also considered to be an important factor affecting the soil URE activity of the AS site. The significantly lower CAT value at the NF site compared to other sites may be due to the restrictive effect of excessive soil water content on microbial growth, which exceeded the activation of CAT activities (Hallaj et al., 2022; Gui et al., 2023). Soil invertase closely participates in the circulation of organic carbon, and soil invertase activity can reflect the soil’s capacity to transform organic carbon (Hu et al., 2023). Soil polyphenol oxidase can degrade phenols produced by plant lignin and detoxify soil (Toscano, Colarieti & Greco, 2003). The INV and PPO values were significantly higher at the VC, FB, TB, and SS sites than at the AS and NF sites. These results indicate that the artificial vegetation eco-restoration measures discussed in this study improved soil remediation and the soil’s capacity to transform organic carbon. The soil biological metabolic activity at the AS site was not exuberant, with low humus, which may have resulted in low organic carbon transformation efficiency and a low INV value. Research has shown that polyphenol oxidase activity is relevant to soil oxygenation and moisture status (Ankegowda et al., 2022). Compared to the AS site, the artificial vegetation eco-restoration sites and the NF site in this study were both considered oxygen and water sufficient. The reason INV and PPO values of the NF site were significantly lower than the artificial vegetation eco-restoration sites may be because the acidic soil environment was unfavorable for the catalyzed reaction of these two enzymes (Lin et al., 2022).

The MBN values at the FB and VC sites were lower than those at other sites and showed no significant difference. This result suggests that the FB and VC sites had low levels of nitrogen transformation and utilization, making soil microbial reproduction slower at these two sites. Though MBP has a longer transformation than MBN, MBP is an extremely active component in the soil organic phosphorus fraction (Sun et al., 2013). The MBP results showed that phosphorus transformation level was one of the main factors affecting SQ at the FB site. On the whole, soil microbial biomass carbon content at the four artificial vegetation eco-restoration sites was lower than at the NF site. Although the soil organic matter at artificial vegetation eco-restoration sites was found to be at a medium level or slightly above, the activity of soil organic carbon at these sites need to be improved (Khadem et al., 2021).

Comparison of soil quality indexing methods

The total data set containing soil physical, chemical, and biological indicators can more comprehensively present the results of the soil quality assessment, but with a heavy computational burden, and the correlations among these indicators may cause information overlap (Fig. 3). The minimum data set, which reduces the workload of the soil indicator measurements without losing soil quality information, is widely used for soil indicator selection when assessing the soil quality of different land uses and restoration treatments (Raiesi & Pejman, 2021; Zhou et al., 2020; Lu et al., 2023). However, this approach may exclude the limiting soil indicators of some sampling sites when selecting soil indicators from the total data set by PCA (Liu et al., 2018). Some important soil indicators representing the soil physical, chemical, or biological properties may be omitted just because they were not highly weighted in any of the selected individual PCs (Yu et al., 2018). Therefore, a revised minimum data set approach was adopted in this article to ensure the soil indicators contained in the minimum data set represented the soil biological, chemical, and physical properties. Three biological soil indicators (CAT, MBP, and URE), three chemical soil indicators (AN, TK, and AK) and one physical soil indicator (BD) were retained in the revised minimum data set, whereas only four biological soil indicators (PPO, URE, MBP, and INV) and one chemical soil indicator (AN) were retained in the minimum data set (Table 5). The revised minimum data set equally and comprehensively considered all soil properties, thereby improving the discrimination of the weighted additive soil indicator and highlighting the importance of all soil properties (Yu et al., 2018).

According to the correlation and regression analysis (Figs. 5, 6), the accuracy of SQI values based on the revised minimum data set was superior to the accuracy of SQI values based on the minimum data set. The Nash efficiency coefficient ( Ef) and the relative deviation coefficient ( ER) were then calculated to compare the accuracy of SQI values based on MDS and RMDS (Lin et al., 2023). The Ef value of SQI-NLRM was higher and the ER value of SQI-NLRM was lower than that of the other soil indicator data set approaches (Table 6). These results indicate that nonlinear scoring based on the revised minimum data set (SQI-NLRM) can distinguish the influence of vegetation eco-restoration measures at the Xiangjiaba hydropower station on soil quality more sensitively and accurately than other methods.

Table 6 The accuracy assessment of SQI value based on MDS and RMDS.

	SQI-LM	SQI-NLM	SQI-LRM	SQI-NLRM	
Ef	0.162	0.538	0.940	0.962	
ER	0.149	0.073	0.020	0.013	

Evaluation of soil quality under different vegetation eco-restoration techniques

Vegetation eco-restoration techniques significantly affected soil quality in the disturbance areas at the Xiangjiaba hydropower station. SQI values differed significantly between the artificial vegetation eco-restoration sites and the unrestored disturbance site (Fig. 4). Using the soil quality graded standard established by many scholars on different land use types, such as mining wastelands and soil tillage management, the soil quality of disturbed areas at the Xiangjiaba hydropower station could be divided into five groups, as shown in Fig. 7 (Zhao et al., 2020). Based on the SQI-NLRM, high SQI values were observed at the TB, VC, and NF sites, with a medium SQI value observed at the FB. The artificial restoration base materials of the TB and VC sites have strong anti-erosion abilities. The dominant species of these two sites belong to Leguminosae (Zhao et al., 2021). The plant root of Leguminosae can secrete large amounts of sugars and amino acids, resulting in a positive impact on soil microorganism and nutrients (Yao et al., 2022). As a result of the effect of these factors, the SQI values of the TB and VC sites were greater than the SQI values of other sites. The trend in the range of values for SQI-NLRM was similar to that of SQI-NLT, indicating that SQI-NLRM is suitable for accurately assessing the soil quality of vegetation eco-restoration sites.

Figure 7 Soil quality grades under different vegetation eco-restoration modes.

(A) SQI-NLRM, (B) SQI-NLT.

The applicability of the revised data set for soil quality assessment under different vegetation eco-restoration techniques at engineering disturbed areas was proven in this study. The SQI calculated based on RMDS reflects the combined response of seven important indicators to different vegetation eco-restoration measures: CAT, MBP, URE, AN, TK, AK, and BD. The contribution ratios of these seven soil indicators contained in the revised minimum data set to the soil quality index were then calculated to analyze the individual indicator response to different vegetation eco-restoration measures (Fig. 8). The contribution of the seven soil indicators significantly varied among different vegetation eco-restoration measures (Fig. 8). The contribution ratio of BD to the SQI of different sites ranged from 30.20% to 46.04%, with an average contribution ratio of 37%, which was the largest contribution indicator to SQI in the study area (Fig. 8). Wang et al. (2023b) found that SOM contributed the most to SQI when evaluating soil quality in grassland and Li et al. (2019) found that alkali-hydrolyzable nitrogen contributed the most to SQI in highly productive farmland during the wheat phase of a wheat-maize cropping system. However, in this study, BD was identified as the largest contributor to SQI. This difference may be attributed to variations in plant types, as the previous studies focused on restoring grass and wheat, and the vegetation community of the study area in the present study was more complex after a long period of ecological restoration, with a combination of trees, shrubs, and grasses. This vegetation community led to an increase in rooting and microbial activity, resulting in significant changes in BD and an improvement in soil structure (Nyirenda, 2020; Gu et al., 2019). Soil bulk density is an important and sensitive soil quality indicator and is closely connected with soil functions, such as the soil nutrient cycle, soil nutrient storage, and soil aggregate degree (Xie et al., 2017). Soil bulk density also has a significant effect on plant root growth and physiological activity (Zhang et al., 2023a). Relatively high bulk density is considered to be the reason why the SQI of the AS site was much lower than the SQI of other sites. The susceptibility of soil bulk density to different vegetation eco-restoration measures and its high contribution ratio to soil quality in disturbed areas of Xiangjiaba hydropower station indicate that soil bulk density is an indicator that can potentially reflect effect of vegetation restoration measures on soil quality. More attention should be paid to soil bulk density in vegetation eco-restoration work.

Figure 8 The contribution of the revised minimum data set indicators to the soil quality index.

The main objective of soil quality assessments is to accurately investigate the condition of the soil, find out whether problems exist, and then provide the basis for soil quality improvement. In engineering applications, the total data set can comprehensively express soil quality information of the study area and has referential value for accurate understanding of the soil quality of study area, but the total data set usually contains multiple indicators that have no ability to reflect or control the effects of vegetation restoration techniques. The soil indicators retained in the RMDS soil quality evaluation method can comprehensively express soil quality information of the study area, and the evaluation results can reflect the impacts of different vegetation eco-restoration measures on soil quality. The RMDS is recommended for assessing soil quality during vegetation eco-restoration in disturbed areas, such as the Xiangjiaba hydropower station and other areas with similar habitat characteristics. Simplifying the total data set can effectively and accurately guide the vegetation restoration effect monitoring and control work of disturbed areas. However, the selected soil quality evaluation method should depend on the actual requirements and purpose of the evaluation in engineering practice. Additionally, given the potential influence of seasonal variations and meteorological conditions on plant growth and the outcomes of ecological restoration, temperature and precipitation should be further examined as variables in future studies.

Conclusions

This study compared six SQIs calculated using various scoring functions and weighting methods and found that all six were significantly positively correlated with one another (P < 0.01). These results indicate that the six SQIs are equally sensitive and accurate, and can represent soil quality and quantify the effects of vegetation eco-restoration techniques on soil quality. They reflect a higher ability to combine and synthesize the complicated information contained in the total data set, SQI-NLRM is suggested to be a more effective method for detecting the influence of vegetation eco-restoration techniques on soil quality than other methods. The results show that four vegetation eco-restoration techniques (vegetation concrete eco-restoration slope, frame beam filling soil slope, thick layer base material spraying slope, and external-soil spray seeding slope) have positive effects on soil quality in the disturbed areas at the Xiangjiaba hydropower station. Application of artificial remediation measures may accelerate soil quality restoration and repair disturbed ecosystems in engineering disturbed regions. SQI based on RMDS has important application value for regional scales because it can be used to integrate complicated information from a large data set, and measuring methods for soil indicators contained in the RMDS are conventional soil experiments. It is recommended that these seven soil indicators (CAT, MBP, URE, AN, TK, AK, and BD), especially soil bulk density, should be included in the engineering disturbed or vegetation restoration soil quality comprehensive assessment database of the study area. SQI-NLRM is an effective tool for soil quality assessment and management with simple calculations and high accuracy and is recommended for soil quality assessment at other areas with similar habitat characteristics.

Supplemental Information

Supplemental Information 1 Raw data.

Supplemental Information 2 Evaluate six vegetation ecological restoration sample plots using six soil quality evaluation methods and compare which evaluation method is more suitable for ecological restoration of disturbed areas.

Photo credit: Xia Dong

Supplemental Information 3 Measurements of AK at each study site.

Supplemental Information 4 Measurements of AN at each study site.

Supplemental Information 5 Measurements of BD at each study site.

Supplemental Information 6 Measurements of CAT at each study site.

Supplemental Information 7 Measurements of INV at each study site.

Supplemental Information 8 Measurements of MBC at each study site.

Supplemental Information 9 Measurements of MBN at each study site.

Supplemental Information 10 Measurements of MBP at each study site.

Supplemental Information 11 Measurements of pH at each study site.

Supplemental Information 12 Measurements of POR at each study site.

Supplemental Information 13 Measurements of PPO at each study site.

Supplemental Information 14 Measurements of SOM at each study site.

Supplemental Information 15 Measurements of TK at each study site.

Supplemental Information 16 Measurements of TN at each study site.

Supplemental Information 17 Measurements of URE at each study site.

Supplemental Information 18 Measurements of WAT at each study site.

Supplemental Information 19 Measurements of soil indicators at each study site.

Additional Information and Declarations

Competing Interests

Author Contributions

Data Availability

The authors declare that they have no competing interests.

Bingqin Zhao conceived and designed the experiments, analyzed the data, prepared figures and/or tables, authored or reviewed drafts of the article, and approved the final draft.

Ruzhang Gao performed the experiments, analyzed the data, prepared figures and/or tables, and approved the final draft.

Xingfeng Zhang performed the experiments, prepared figures and/or tables, and approved the final draft.

Lu Xia performed the experiments, prepared figures and/or tables, and approved the final draft.

Lun Zhang conceived and designed the experiments, authored or reviewed drafts of the article, and approved the final draft.

Dong Xia conceived and designed the experiments, authored or reviewed drafts of the article, and approved the final draft.

Daxiang Liu conceived and designed the experiments, authored or reviewed drafts of the article, and approved the final draft.

Zhenyao Xia conceived and designed the experiments, authored or reviewed drafts of the article, and approved the final draft.

Wennian Xu conceived and designed the experiments, authored or reviewed drafts of the article, and approved the final draft.

The following information was supplied regarding data availability:

The raw measurements are available in the Supplemental File.

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
