# Peer review of "Comparison of soil quality assessment methods for different vegetation eco-restoration techniques at engineering disturbed areas"

_PeerJ, doi:10.7717/peerj.18033_

## Round 0.1 · original submission · Major Revisions

Kindly incorporate the feedback provided by all reviewers and resubmit the revised manuscript along with a point-by-point rebuttal letter. Additionally, in your revision, please emphasize the novelty of the manuscript and its potential for universal application or validation in other contexts. Furthermore, please ensure that the language of the paper is refined to enhance clarity and coherence.

Reviewer 1 ·

Basic reporting

Restoration of vegetation ecosystems in disturbed area is a major issue for scientists and establishment of an efficient and accurate quantitative measurement on soil quality is one of the priorities. Zhao et al. compared the different soil quality assessment methods on real-world dataset collected at Xiangjiaba hydropower station and found SQI-NLRM is the better method among all. The manuscript is overall well-structured and well-written, and would provide valuable data points and insights to this field, thus I would support the publication of this work in PeerJ if the authors could address some concerns below.

Overall, there is an abuse of abbreviation in the manuscript, and it would be better to spell it out when possible. Example includes: Table-3, Table-4, Figure-1, Figure-5 (N & NL: Linear & Non-linear)

Background: The authors provide sufficient background for different soil assessment methods. However, more background or explanation regarding the differences and known facts about the natural eco-restoration (NF) and 5 artificial remediation techniques (VC, FB, TB, SS, AS) is still needed.

Line-50 to 92: The authors exhaustively listed current studies of soil assessment methods. I suggest the authors to make a table to summarize the pros and cons as comparison rather than just listing here

Line-221/Table-4: The authors should also show the PCA plot (at least PC1 and PC2) and highlight how different parameters contribute the difference.

Figure-2: make font size of number smaller to avoid overlap

Figure-3: (a) I would ask the authors to replot Figure-3 similar as Figure-6 (different techniques will have different color or separate plot) to highlight the correlation between methods and difference between techniques. Current plot is a little confusing in terms of comparison. (b) Please explain the letter above the bar: a, b, c, d

Figure-4: please change it into a heatmap with values as all correlation values in this plot is pretty high.

Experimental design

no comment

Validity of the findings

no comment

Additional comments

no comment

·

Basic reporting

1. Consult a fluent English speaker to refine the language, grammar, and writing style, ensuring optimal clarity and coherence.
2. Visualize the result in Table 3 as a graph. A graphical representation would enhance comprehensibility. Additionally, relocate Table 3 to the supplementary material section.

Experimental design

1. Is it feasible to replicate or triplicate the sample within a 3-day period? Alternatively, is it possible to incorporate temperature and precipitation as additional parameters? Seasonal variations and meteorological conditions might influence the outcome and plant growth.
2. Provide the Principal Component Analysis (PCA) plot.
3. Expand the discussion and compare the final six Soil Quality Indices (SQIs) with other publications, highlighting the differences in vegetation types. Draw parallels and contrasts with existing literature.

Validity of the findings

I am uncertain whether the results obtained can be universally applied or validated in other locations with consistent outcomes.

Reviewer 3 ·

Basic reporting

1. The text of the article is not very clear and requires a better choice of words.
2. The manuscript is based on an appropriate review of existing work in both the introduction and the discussion sections.
3. The manuscript is appropriately structured and supported by figures and tables.

Certain sections have been marked in the text that the authors may consider rephrasing to create a more impactful presentation of their ideas. They may seek the help of a professional language editor for the purpose.

Experimental design

The methods section distinctly elucidates the questions being asked and the need for answering these questions.
The methodology described and the statistical analysis carried out adequately address the questions being asked and promore impactful presentation of their ideas. ide the necessary answers.

Validity of the findings

The study validates its findings and has explicit conclusions that can be used to plan and replicate similar interventions in engineering disturbed landscapes.

Additional comments

The findings of the study are of great relevance in amelioration of effects of disturbances in landscapes caused by infrastructure development.

Annotated reviews are not available for download in order to protect the identity of reviewers who chose to remain anonymous.

Reviewer 4 ·

Basic reporting

Comparison of soil quality assessment methods for different vegetation eco-restoration techniques at engineering disturbed areas

Minor Revision
Overall, this paper is a valuable study that looks at different ways to check how healthy soil is when we're trying to fix areas where humans have messed up the environment. It's an important topic, and the research is done well. Here are a few small ideas to make the paper easier to understand and more complete

• Clarify the rationale behind the selection of vegetation eco-restoration techniques for comparison to provide context for the study's significance.
• Ensure that the methodology section includes specific criteria used for selecting the vegetation eco-restoration techniques, enhancing transparency and reproducibility.
• Organize the results section in a structured manner, facilitating easier comparison by grouping results according to vegetation eco-restoration technique or soil quality parameter.
• Expand the discussion to include implications of the findings for soil management practices in engineering disturbed areas, providing deeper insights into decision-making processes for ecosystem restoration efforts.
• Incorporate a brief discussion of study limitations and potential avenues for future research in the conclusion, guiding future investigations in the field.
• Additionally, ensure consistency in terminology and grammar throughout the manuscript for clarity and readability.

Experimental design

n\a

Validity of the findings

n\a

Additional comments

n\a

---

## Round 0.2 · accepted · Accept

All comments are duly incorporated. So, the paper is accepted.

Reviewer 1 ·

Basic reporting

I am glad to see the authors have address all of my concern and I would support the publication of the revised manuscript.

Experimental design

no comment.

Validity of the findings

no comment.

Additional comments

no comment.

·

Basic reporting

The manuscript now reads much more fluently, with enhanced clarity in the articulation of the research objectives, methodology, and findings. The revisions have effectively reduced ambiguities and improved the precision of the scientific arguments, making the content more accessible and understandable to the readership.

Experimental design

no comment

Validity of the findings

no comment

Reviewer 3 ·

Basic reporting

The Language has been improved and the document is much easier to understand.

Experimental design

No comment

Validity of the findings

The document has been much improved and the conclusions are based on the results provided.

Additional comments

The Authors have put in effort and the modified manuscript may be considered of publication.

Reviewer 4 ·

Basic reporting

All comments are resolved and no further comments from my side and it suggest to accept it as it is.

Experimental design

n/a

Validity of the findings

n/a

Additional comments

n/a